# Coupled Thermodynamics and Phase Diagram Analysis of Gas-Duct Concretion Formation in Pyro-Processing Ironmaking and Steelmaking Dust

**Daya Wang [1,2], Shaoguang Hua [1,2], Liushun Wu [3,*], Kunlong Liu [3] and Haichuan Wang [3]**

[1] Huawei National Engineering Research Center of High Efficient Cyclic Use of Metallic Mineral Resources Co., Ltd., Maanshan 243071, China; dayawangustb@126.com (D.W.); huasg@sinosteel.com (S.H.)

[2] Sinosteel Maanshan Institute of Mining Research Co., Ltd., Maanshan 243071, China

[3] School of Metallurgical Engineering, Anhui University of Technology, Maanshan 243002, China; ahutlkl@ahut.edu.cn (K.L.); which@ahut.edu.cn (H.W.)

* Correspondence: wuliushun@ahut.edu.cn; Tel.: +86-18155581066

**Abstract:** In recent years, the steel industry has accumulated approximately 100 million tons of dust annually, severely threatening the environment. Rotary kiln technology is one of the main industrial methods used to process this dust. However, some substances in flue gas congeal on the cooling wall of the gas duct and seriously affect production. In this study, the properties and formation mechanisms of the coagulum were investigated on the basis of experimental and thermodynamic analyses. The experimental results showed that the coagulum is mainly composed of chlorides (KCl, NaCl, and $ZnCl_2$), oxides (ZnO, FeO), and carbon, with three structures: lumps, fibers, and particles. Based on a thermodynamic analysis, a reasonable explanation was proposed to clarify the formation mechanism. The liquid phase (a eutectic system of KCl–NaCl–$ZnCl_2$), dendrites (KCl, NaCl), and particles (ZnO, FeO, C) were found to act as binders, stiffeners, and aggregates in the coagulum, respectively, constituting a composite structure. Liquids acting as binders are essential for coagulum formation, and dendrites and particles strengthen this effect. Furthermore, the eutectic system of chlorides plays a crucial role in coagulum formation. The results of the present study offer a theoretical understanding of gas-duct coagulation and will provide guidance for adopting alleviation measures.

**Keywords:** dusts from steel manufacturing; coagulation; thermodynamics analysis; phase diagram; rotary kiln; chlorides

## 1. Introduction

Approximately 1.8 billion tons of crude steel were produced worldwide in 2020. This amount corresponded to more than 100 million tons of dust, accounting for approximately 8%–12% of the crude steel output. Dust containing heavy metals and cyanide is classified as hazardous waste [1–3], and poses a threat to the environment. To reduce resource consumption and environmental pollution, such dust has been used as raw materials for blast furnace ironmaking [4–6]. However, the use of this dust causes the accumulation of zinc and alkali metals in the blast furnace, shortening the service life of the blast furnace and leading to poor production [7]. In recent years, some researchers have proposed new methods based on two schemes: integral use and recycling metals.

Figure 1 shows a graphic illustration of the methods based on integral use. Dust with a high specific surface area and high iron and carbon contents has previously been applied to treat wastewater, such as in demulsification [8]; to absorb heavy metals [9]; and to facilitate catalysis in organic degradation [10,11], anaerobic digestion [12,13], and biomass pyrolysis [14]. These applications provide added value to the dust. However, dust used for these purposes also requires subsequent treatment. In addition, dust is also used to directly produce materials such as glaze [15] and mortars [16–18].

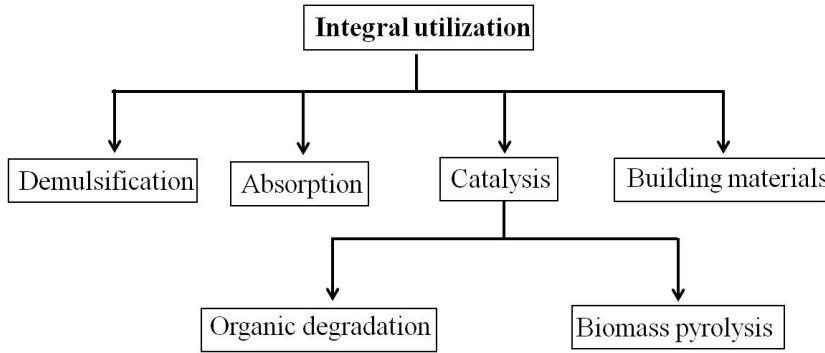

**Figure 1.** Graphic illustration of the integral use methods for dust.

The methods for recycling metals are illustrated in Figure 2. The metals in dust are recycled through wet leaching or pyro-separation. Wet leaching can be divided into two types: the nonselective and selective leaching of zinc. In nonselective leaching, sulfuric acid is used as the leaching reagent [19,20]. During leaching, several oxides in the dust react with the acid and dissolve in the solution. In particular, a large amount of iron enters the solution, with no beneficial acid consumption. The methods used for selective leaching can be divided into acid leaching and alkali leaching. Iminodiacetic acid [21] or enoic acid [22] is adopted in the acid leaching method, and ammonia water [23] or an NaOH solution [24] is used in the alkaline method to selectively leach zinc. The complex state of zinc in dust results in relatively low leaching efficiency. Moreover, the disposal of wastewater and residue is an additional challenge in wet leaching.

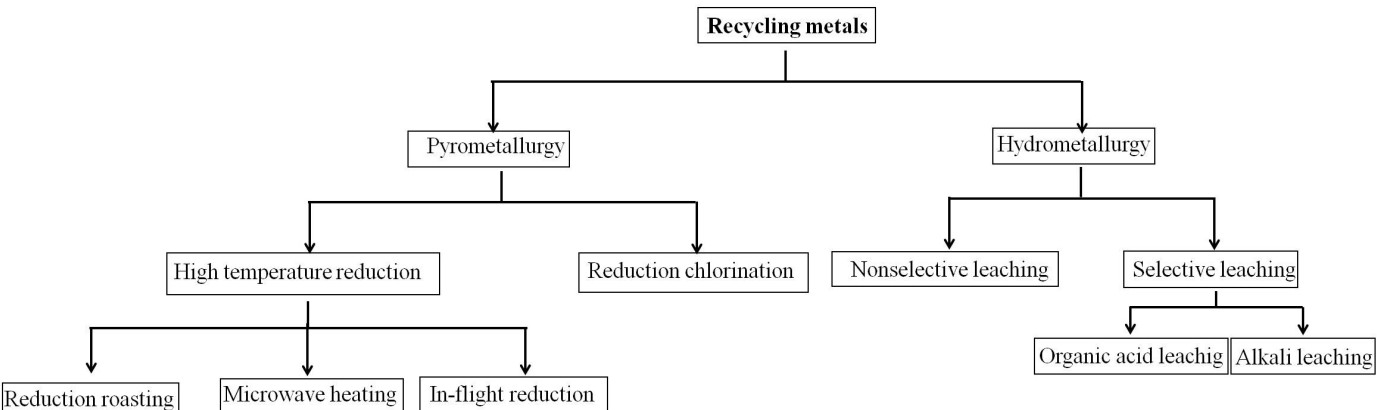

**Figure 2.** Graphic illustration of the methods for recovering metal from dust.

Pyro-separation can be divided into two categories: chlorination roasting and high-temperature reduction. In the chlorination-roasting method [25], ash is mixed with $FeCl_2$ waste liquid, which is then dried and roasted. During roasting, some metals, such as zinc and lead, form chloride vapor, which escapes with the flue gas. Then, chloride is separated from the flue gas by cooling. Roasted materials are treated as raw materials in ironmaking and steelmaking. In high-temperature reduction methods, dust is treated via in-flight reduction technology [26], microwave heating [27,28], and roasting in rotary kilns [1,2] or rotary hearth furnaces [1,29]. A large number of oxides in dust are reduced, and the products with lower boiling points, such as zinc and lead, form metal vapor and escape with the flue gas. The flue gas is then cooled, and crude zinc powder is obtained using a bag-type dust remover. The roasted products are selected as raw materials for iron and steel smelting.

Currently, the method of reduction roasting in rotary kilns or rotary hearth furnaces is industrialized, achieving high separation efficiency for zinc. The substances in flue gas

continually condense on the cooling wall of the heat exchanger in the process, forming a concretion. This concretion blocks the flue gas, which seriously affects smooth production. Although several measures have been taken to alleviate this issue, little attention has been given to the causes of caking. In this study, the mechanism of coagulum formation was experimentally and thermodynamically analyzed, providing theoretical guidance for alleviating concretion.

## 2. Materials and Methods

Metallurgical dust is processed primarily using a rotary kiln or rotary hearth furnace. The rotary kiln process has greater adaptability to feedstock than the rotary hearth furnace process. Hence, this study focused on the rotary kiln process (Meishan iron & steel company Ltd., Nanjing, China), which is schematically shown in Figure 3. As shown, in this method, solid feedstock enters the kiln from the inlet and then passes through a drying region (650–700 °C), a preheating zone (700–850 °C), and a high-temperature section (850–1200 °C). Finally, the feedstock exits from the furnace end (at approximately 1100 °C, with approximately 1% surplus carbon in the clinker). The gas traverses the kiln in the opposite direction of the solid material. In the high-temperature section, some of the oxides in the solid are reduced, and some volatile metals and compounds enter the gas phase. The gas exits from the kiln inlet (550–850 °C), passes through the settling chamber followed by the heat exchanger (inlet temperature: approximately 300 °C), is cooled to <160 °C, and subsequently passes through a filter-bag dust-removal system, reaching a temperature of approximately 60 °C.

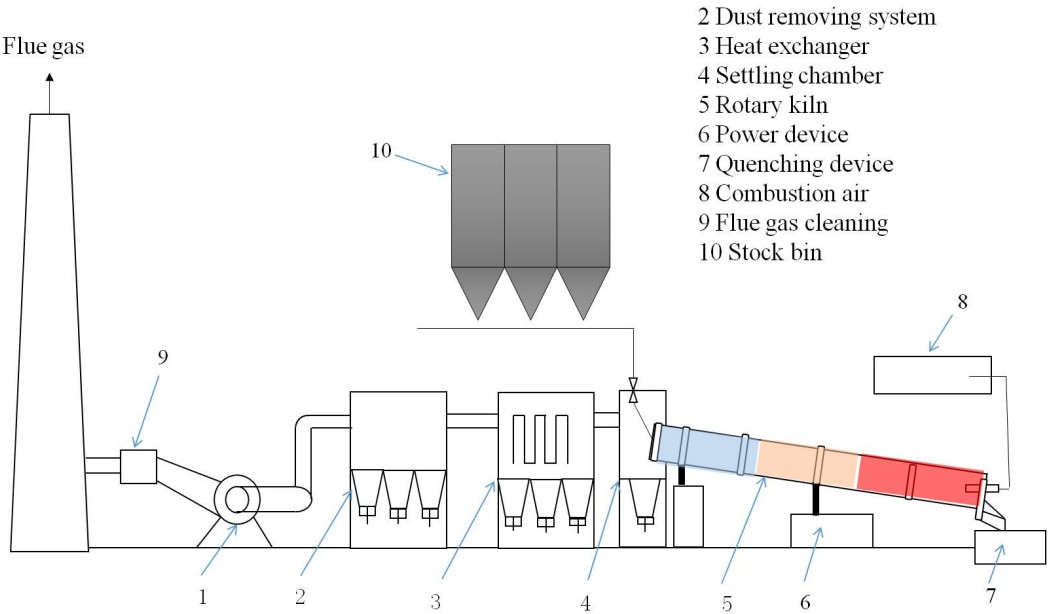

**Figure 3.** Processing flow of metallurgical dust using a rotary kiln (blue, orange and red represent drying region, preheating zone, and a high-temperature section.

When flue gas passes through the heat exchanger, some substances in the gas coagulate on the cooling wall of the heat exchanger (Section 3 in Figure 3). Crude zinc powder is obtained during bag dusting (dust removing system). Because of the differences in the heating methods employed and the nature of the feedstock, the ratio of the gas volume to the clinker weight ranges from 2 to 5 $m^3$/kg. The flue gas mainly contains $N_2$ (68%–70%), $CO_2$ (18%–20%), water vapor (10%–11%), and $O_2$ (1%–2%) (analyzed by gas chromatography, GC 2010 PLUS, Shimadzu, Kyoto, Japan).

In the present study, mixtures of metallurgical dusts (such as blast-furnace dust and electric-arc-furnace dust) and other metallurgical solid wastes (for example, steel pickling sludge) were treated in a rotary kiln. The coagulum blocked flue-gas flow, which suspended operation at 100–200-day intervals. After maintenance, the next period was started.

We investigated two production cycles that lasted 114 (case 1) and 151 (case 2) days. In each cycle, the feedstock and crude zinc powder were sampled once every 10 days. During each shutdown, five coagulum blocks were obtained from different locations of the heat exchanger along the gas inlet to the outlet. Each block was cut into three pieces perpendicular to the thickness of the coagulum, and each piece was cut into two blocks parallel to the thickness direction. One block was smoothed and polished. The samples were prepared without a cleaning agent to prevent a reaction, and cleaning was performed using a dryer. Another block was first crushed in a vibration mill and then powdered in a ball grinder for 5 h.

The three powders of the feedstock, coagulum, and crude zinc powder were dried in an oven for 24 h at 110 °C. The dried powders were then ground in a ball grinder for 5 h. The chemical compositions of the dried powders were determined by X-ray fluorescence (ARL Advant'X IntelliPower 3600, Thermo Fisher Scientific, Waltham, MA, USA), precision: ±5%). The phases in the three powders (about a 5 g sample each) were analyzed by X-ray diffractometry (Bruker D8 Advance, Bruker, Karlsruhe, Germany, Cu K) with a step size of 0.01 degree per 0.3 s. Phase percentages were determined by the k-value method using silica as a reference. Chemical compounds in the coagulum were determined using a Raman spectrometer (inVia, Gloucester, UK). The thermal properties of the three powders were analyzed at a heating rate of 10 K/min in an air atmosphere using a thermal analyzer (DTG-60H, Shimadzu, Kyoto, Japan, precision: ±1 °C and ±1%). The morphology of the coagulum block sample was observed using a scanning electron microscope (SEM, JSM-6510, JEOL, Tokyo, Japan), and the compositions of the phases in the coagulum were determined using energy dispersive spectroscopy (EDS, INCA Feature X-Max 20, Oxford Instruments, Oxford, UK, precision: ±5%).

## 3. Results

Figure 4 shows images of the feedstock of the rotary kiln, the coagulum on the wall of the heat exchanger, and the crude zinc powder collected using a bag-dust removal device. The contents of the main elements (except for oxygen and carbon) in the feedstock, the coagulum, and the crude zinc powder are listed in Table 1.

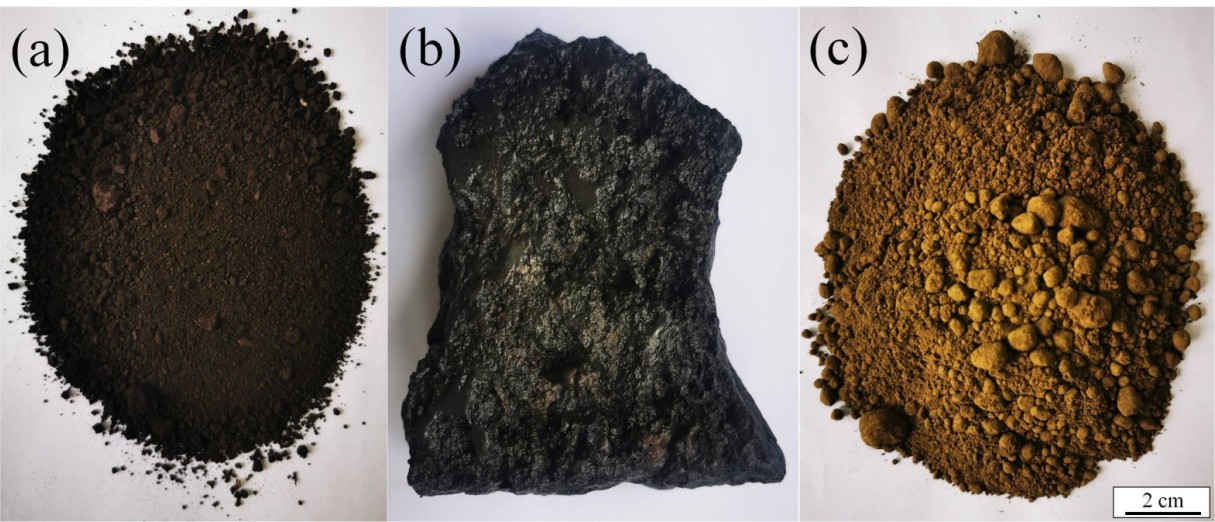

**Figure 4.** Images of feedstock (**a**), coagulum (**b**), and crude zinc powder (**c**).

**Table 1.** Element contents in the feedstock, coagulum, and crude zinc powder (mass %).

| Element | Fe | Cl | K | Zn | Ca | Na | Pb | Si |
|---|---|---|---|---|---|---|---|---|
| Feedstock | 30–35 | 6–10 | 5–7 | 5–8 | 4–6 | 1–6 | <1 | 2–5 |
| Coagulum | 9–12 | 25–28 | 9–12 | 20–23 | <1 | 2–7 | 4–5 | <1 |
| Crude zinc | 5–6 | 24–26 | 7–9 | 35–38 | <0.5 | 4–6 | 3–4 | <1 |

Figure 5 shows the XRD patterns of the feedstock, coagulum, and crude zinc powder. As shown, the feedstock mainly contained $Fe_2O_3$, $SiO_2$, and carbon; the coagulum mainly contained carbon, salts (KCl, NaCl, $ZnCl_2$), and oxides (FeO, ZnO); and the crude zinc powder was composed of ZnO, $K_2ZnCl_4$, NaCl, and $KFeO_2$.

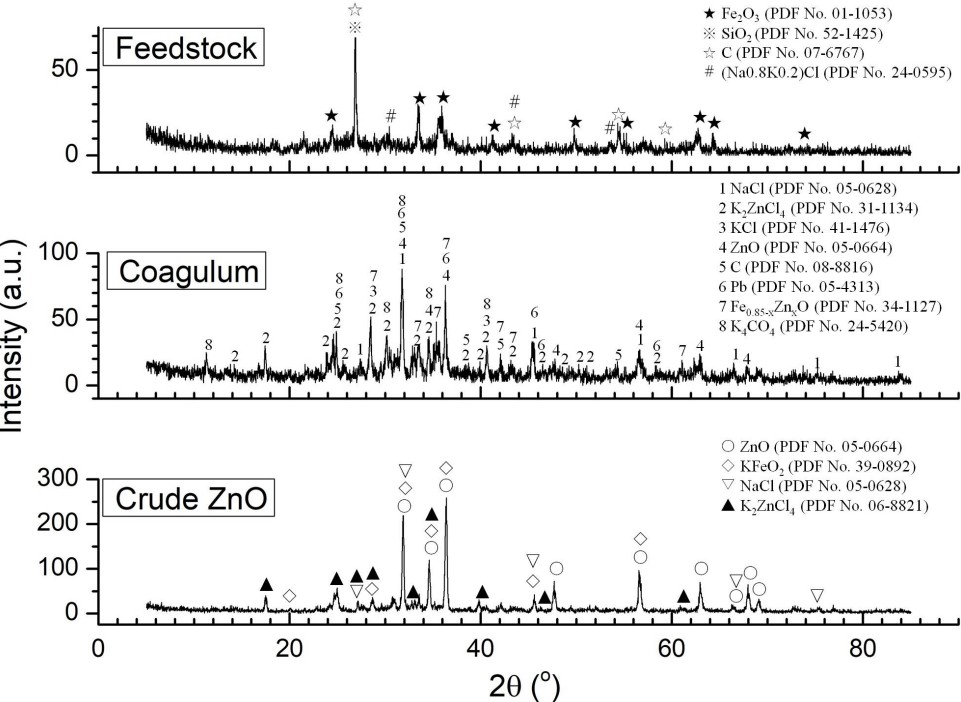

**Figure 5.** XRD patterns of feedstock, coagulum, and crude zinc powder.

Based on our XRD experiments using the k-value method, we conducted qualitative calculations. Moreover, equilibrium calculations were performed according to the chemical compositions of the samples. Using these two techniques, the average contents of ZnO, FeO, KCl, NaCl, and $ZnCl_2$ were evaluated, with approximate values of 14%, 11%, 17%, 9%, and 18%, respectively.

Figure 6 shows SEM images of the coagulum. The chemical compositions of the different regions in Figure 6 determined by EDS are listed in Table 2. Based on the EDS analysis results, the A region mainly contained carbon; the B region contained carbon, chlorides, and oxides; the C region was mainly composed of oxides and carbon; the D region consisted of KCl and carbon; the E region contained carbon and chlorides; and the F zone mostly contained carbon and chlorides.

Figure 7 shows the thermogravimetric analysis (TGA) and differential thermal analysis (DTA) results for the coagulum. The weight-loss rate reached 1%, 4%, and 56% at 140 °C, 200 °C, and 970 °C, respectively. Two endothermic peaks were observed at approximately 160 and 360 °C. For the feedstock, the weight loss rate was approximately 40% at 1000 °C. For crude zinc powder, the weight loss rate was approximately 48% at 1000 °C.

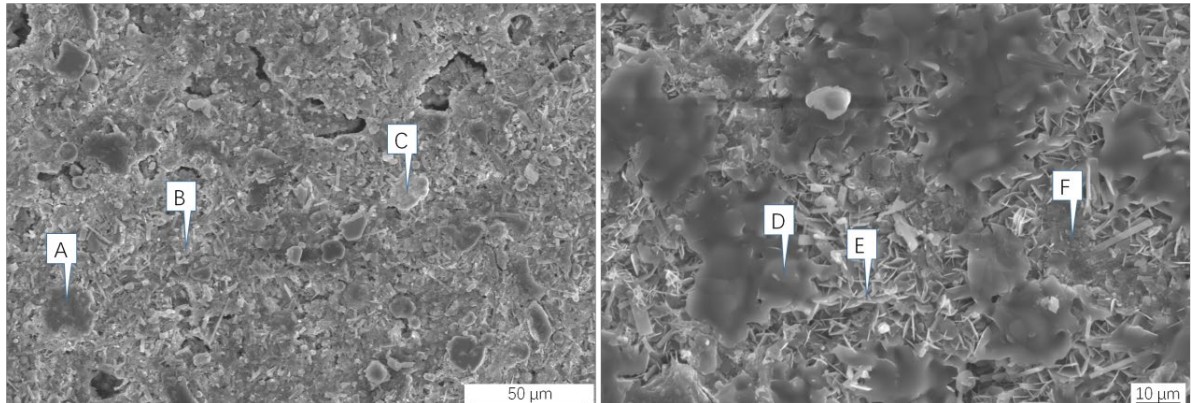

**Figure 6.** SEM images of the coagulum.

**Table 2.** Element composition in marked regions of Figure 6 (mol%).

| Element | C | O | Al | Si | S | Cl | K | Ca | Fe | Zn | Pb |
|---------|-------|-------|------|------|------|-------|-------|------|-------|-------|------|
| A | 91.97 | 5.57 | – | – | 0.43 | 0.46 | 0.28 | 0.25 | 0.3 | 0.74 | – |
| B | 51.13 | 35.64 | – | – | 2.63 | 2.41 | 2.67 | 0.93 | 1.24 | 3.35 | – |
| C | 20.96 | 35.75 | 1.03 | 3.5 | – | 4.69 | 1.72 | – | 17.42 | 14.92 | – |
| D | 26.69 | 11.2 | – | – | – | 30.63 | 26.26 | – | 1.14 | 4.08 | – |
| E | 50.77 | 27.51 | – | – | 0.73 | 8.51 | 3.07 | – | 1.46 | 7.95 | – |
| F | 60.22 | 13.94 | 0.83 | 0.57 | 0.8 | 11.48 | 3.99 | – | 3.21 | 4.68 | 0.29 |

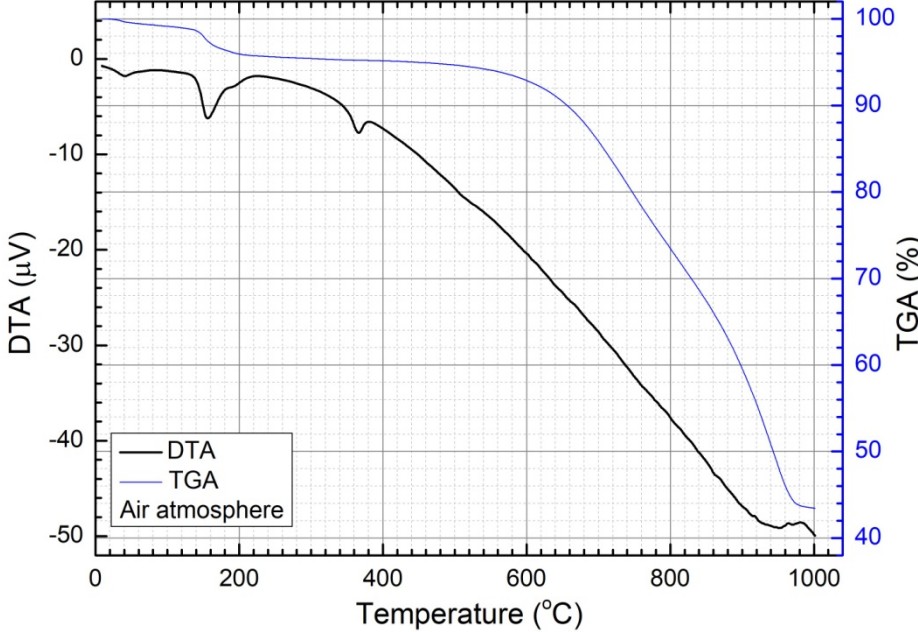

**Figure 7.** Thermogravimetric analysis (TGA) and differential thermal analysis (DTA) for the coagulum.

The Raman spectra of the coagulum are illustrated in Figure 8. The peaks at 1350, 1590, and 2692 cm$^{-1}$ were assigned to graphite [30,31]; those at 220, 291, 333, and 403 cm$^{-1}$ matched the characteristic peaks of $ZnCl_2$–KCl–NaCl [32,33]; those at 566 and 650 cm$^{-1}$ approached the characteristic peaks of ZnO [34]; and those at 983 and 1008 cm$^{-1}$ corresponded to the characteristic peaks of FeO–ZnO [35].

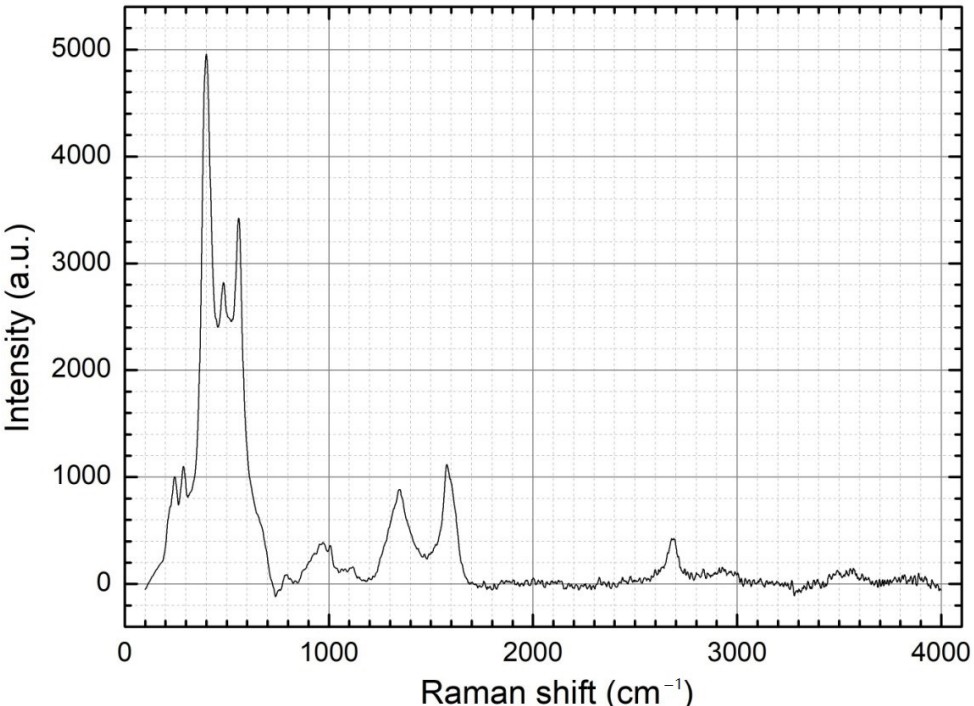

**Figure 8.** Raman spectra of the coagulum.

The thickness of the coagulum from the inlet to outlet of the heat exchanger gradually increased, and then decreased. The two production cycles of 114 and 151 days corresponded to 7.6% and 6.5% average chloride contents in the feedstock, while the contents of the other elements presented minor differences, which indicates that chloride may play an important role in coagulum formation.

The experimental results indicate that the coagulum, containing oxides, chlorides, and carbon, assumed the form of a mixture (a composite material) that consisted of fibers, particles, and blocks. Clearly, the substances of the coagulum originated from flue gas and were further traced to the feedstock. To clarify the formation mechanism of the coagulum, we will discuss the substances in the feedstock that volatilized at high temperature, the substances in the flue gas that dewed or frosted at low temperatures, and the formation process of the coagulum. The thermodynamic data used in the discussion section were obtained from the thermodynamics software FactSage v7.2 (Vendor: ThermFact Ltd., Quebec, QC, Canada).

## 4. Discussion

The feedstock consisted of blast-furnace dust, electric-arc-furnace dust, and steel-pickling sludge. The blast-furnace dust contained sodium chloride and potassium chloride, while the steel-pickling sludge contained ammonium chloride and ferrous chloride. Ammonium chloride and ferrous chloride decomposed in the preheating zone, producing hydrogen chloride, which, in turn, reacted with the oxides. In the high-temperature section, excess carbon in the feedstock led to an extremely high ratio of partial monoxide pressure compared to that of carbon dioxide. Some oxides were directly reduced, producing metals.

During the flue gas transition from high-temperature to low-temperature zones, the reducibility gradually decreased, which oxidized the metals in the flue gas.

### 4.1. Substances in Feedstock Volatilization

Parts of some oxides were also chloridized, forming chlorides such as $NaCl$, $KCl$, $ZnCl_2$, and $PbCl_2$ (see Equations (1)–(4)). The variation in standard Gibbs energies of chlorination depending on temperature is shown in Figure 9. The standard Gibbs energies

were negative in a temperature range of 300–1500 °C, indicating that $Na_2O$, $K_2O$, ZnO, and PbO can be chloridized by hydrogen chloride:

$$Na_2O + 2HCl = 2NaCl + H_2O \tag{1}$$

$$K_2O + 2HCl = 2KCl + H_2O \tag{2}$$

$$ZnO + 2HCl = ZnCl_2 + H_2O \tag{3}$$

$$PbO + 2HCl = PbCl_2 + H_2O \tag{4}$$

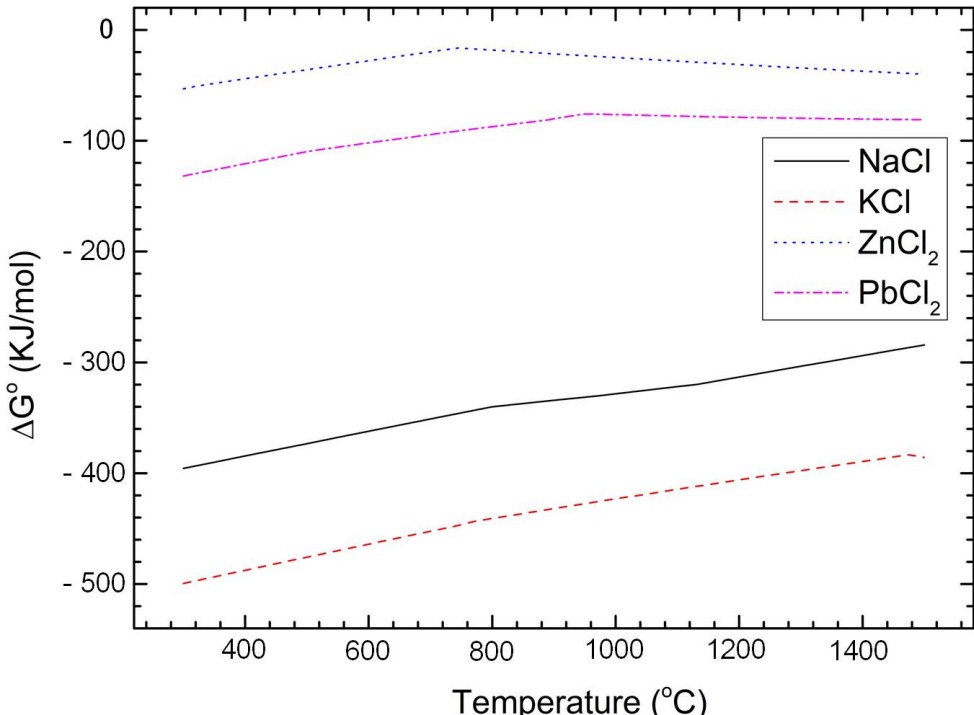

**Figure 9.** Standard Gibbs energies of the chlorination at different temperatures.

$ZnCl_2$ and $PbCl_2$ completely volatilized at 1200 °C due to their low boiling points of 732 and 951 °C, respectively. Although NaCl and KCl have high boiling points of 1516 and 1475 °C, respectively, they correspond to high saturated partial pressures at high temperatures, as shown in Table 3. Thus, the chlorides were vaporized and entered their gas phases at high temperatures. The high-temperature section favored the separation of chlorides from the solid materials.

**Table 3.** Vapor pressures (atm) corresponding to dew temperatures (°C).

| Pressure | 0.1 | 0.01 | 0.001 | $4 \times 10^{-4}$ | $3 \times 10^{-4}$ | $1 \times 10^{-5}$ | $5 \times 10^{-6}$ | $1 \times 10^{-6}$ | $1 \times 10^{-7}$ | $1 \times 10^{-8}$ |
|---|---|---|---|---|---|---|---|---|---|---|
| KCl | 1173 | 971 | 822 | 774 | 761 | 633 | 611 | 564 | 506 | 455 |
| NaCl | 1215 | 1008 | 856 | 806 | 792 | 662 | 639 | 591 | 531 | 478 |
| $ZnCl_2$ | 600 | 494 | 412 | 385 | 376 | 296 | 284 | 257 | 222 | 192 |

In the high-temperature section, the maximum temperature was approximately 1200 °C. Excess carbon led to an extremely high ratio of partial monoxide pressure compared to that of carbon dioxide (see Equations (5)–(7)). In this process, direct reduction occurred, producing various metals. Some metals with low boiling points or high vapor pressures, such as Zn ($T_b$ = 908 °C) and Pb, vaporized and entered the flue gas. The formulae for metal vapor formation along with the standard Gibbs free energy (temperature unit:

Kelvin) were as follows. Based on the calculation results obtained using the expressions for standard Gibbs free energy, the reduction reactions of ZnO and PbO occurred at 952 and 762 °C, respectively:

$$CO_2 + C = 2CO$$
$$\Delta G_5^0 = 171.282 - 0.175T \text{ (kJ/mol)} \tag{5}$$

$$ZnO + C = Zn_{(g)} + CO$$
$$\Delta G_6^0 = 362.686 - 0.296T \text{ (kJ/mol)} \tag{6}$$

$$PbO + C = Pb_{(g)} + CO$$
$$\Delta G_7^0 = 267.964 - 0.259T \text{ (kJ/mol)} \tag{7}$$

In practice, the products of CO and metal vapor were not in a standard state—that is, the partial pressures of CO and metal vapor were all <1. According to the composition of the outlet gas, the partial pressure of carbon monoxide was approximately 0.2 standard atmospheric pressure units (atm). According to the ratio of solid feedstock to gas volume, the partial pressure of zinc steam was <0.02 atm, while that of lead steam was <0.001 atm, which led to lower reduction temperatures. As an example, the Gibbs free energy of Reaction (2) is expressed in Equation (8). According to Equation (8), with a decrease in the partial pressure of zinc steam from 0.1 to 0.001 atm, the reduction temperature decreases from 831 °C to 716 °C when the partial pressure of carbon monoxide is 0.2 atm:

$$\Delta G_2 = \Delta G_2^0 + RT\ln(P_{CO} \cdot P_{Zn}) \tag{8}$$

*4.2. Oxidation and Condensation of Substances in the Flue Gas*

Away from the high-temperature section, due to the suction of air and oxidization caused by the iron oxide in the feedstock, the flue gas exhibited an increasingly stronger oxidation capacity. The metal vapors were gradually oxidized by the carbon dioxide and oxygen in the flue gas. The related formulae, along with expressions of standard Gibbs free energy, are expressed in Equations (9) and (10):

$$Zn_{(g)} + CO_2 = ZnO + CO$$
$$\Delta G_9^0 = -189.417 + 0.119T \text{ (kJ/mol)} \tag{9}$$

$$2Zn_{(g)} + O_2 = 2ZnO$$
$$\Delta G_{10}^0 = -946.688 + 0.414T \text{ (kJ/mol)} \tag{10}$$

In the flue gas, the partial pressure of each gas differed from the standard atmospheric pressure. For Equation (10), Gibbs free energy can be expressed using Equation (11):

$$\Delta G_{10} = \Delta G_{10}^0 + RT\ln\left[1/\left(P_{O_2} \cdot P_{Zn}^2\right)\right] \tag{11}$$

According to Equation (11), the low temperature and high partial pressure of $O_2$ in the gas favored the reactions. The flue gas cooled and contained increasing $O_2$ and $CO_2$ (outlet: 1%–2% $O_2$ and 18%–20% $CO_2$) during marching, which satisfied this requirement.

Meanwhile, with a decrease in temperature, the chlorides in the flue gas condensed at different temperatures. In addition, the condensation temperature of the chloride depended on the partial pressure of chlorides in the flue gas, as shown in Table 3. The high-partial-pressure chlorides in the flue gas, such as NaCl and KCl, condensed at a higher temperature; however, the chlorides with low partial pressure in the flue gas, such as $ZnCl_2$ and $PbCl_2$, frosted at lower temperatures.

*4.3. Formation Mechanism of Coagulum*

Caking on the surface of the cooling wall necessitates the formation of a liquid phase. Based on the properties of the substances in the coagulum, chlorides could play a major role in this process. Among the main chlorides, zinc chloride has a relatively low melting

point [36]. Furthermore, ZnCl₂ and other chlorides form a eutectic system. According to the calculation results of the phase diagram module of FactSage and the available literature, when the molar fraction of ZnCl₂ ranges from 0.46 to 0.87 in a KCl–ZnCl₂ binary system, the salts melt completely at 300 °C. In addition, KCl reacts with ZnCl₂, forming two compounds with low melting points: $K_5Zn_4Cl_{13}$ and $KZn_2Cl_5$ [37]. In the NaCl–ZnCl₂ binary system, NaCl and ZnCl₂ combine to form $Na_2ZnCl_4$ [38], which, along with ZnCl₂, constitutes a eutectic system in the ZnCl₂ fraction range of 0.58–0.88. In this work, we produced a diagram of the KCl–NaCl–ZnCl₂ ternary system and plotted experimental data of the three salts on the diagram. In Figure 10, the isotherm nearest to the vertex of ZnCl₂ corresponds to lower temperatures. Compared to case 2, the locations of the data in case 1 are closer to the vertex of ZnCl₂, which indicates that coagulation was formed easily; the shorter operation time verified this result.

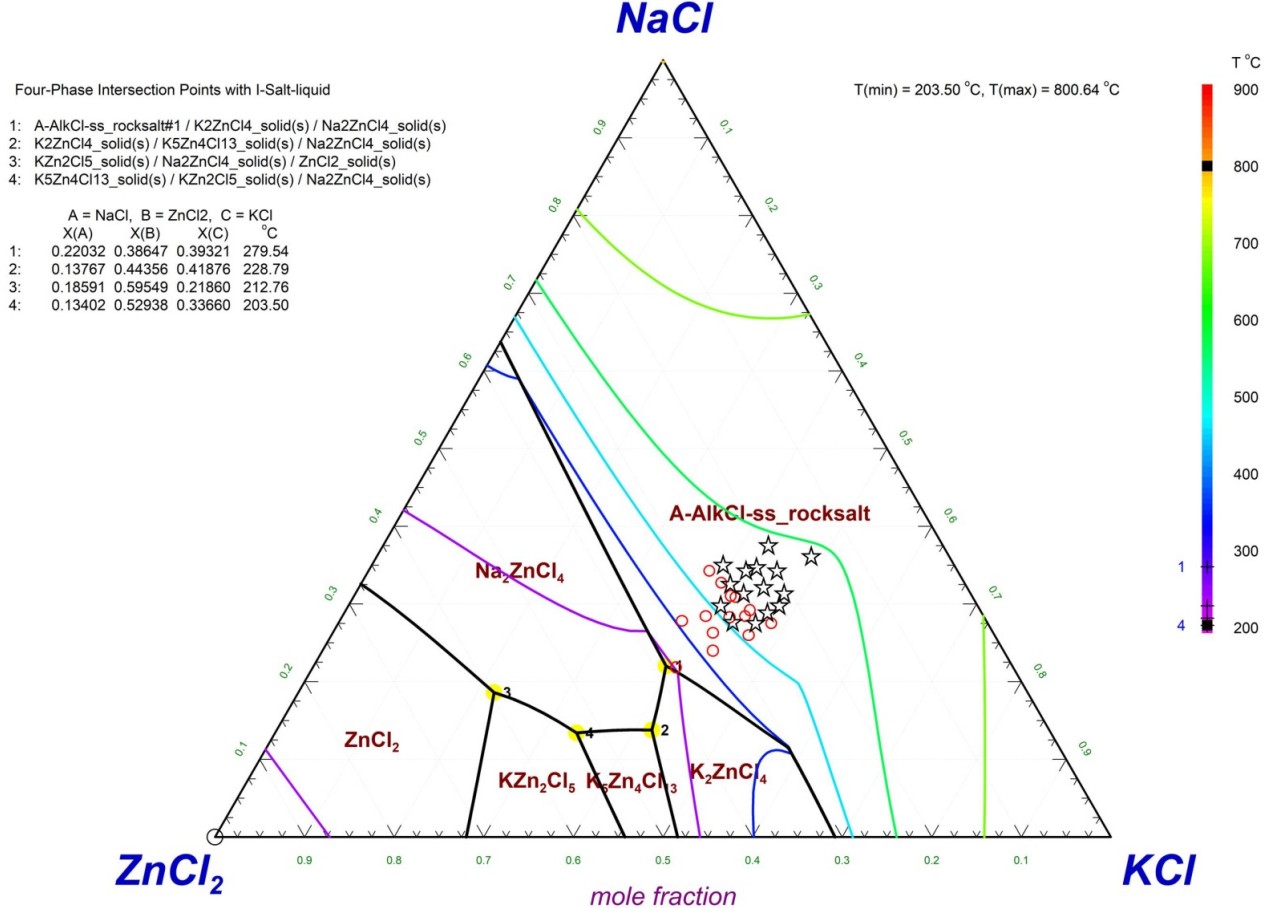

**Figure 10.** Phase diagram of the KCl–NaCl–ZnCl₂ ternary system (case 1: ○, case 2: ☆).

In addition, Robelin et al. [39] showed that KCl–NaCl–FeCl₂ is similar to KCl–NaCl–ZnCl₂ and represents a system with a low melting temperature. The other chlorides, including CaCl₂, FeCl₂, and PbCl₂, further lowered the melting temperature of the salt system, thereby improving coagulation formation.

As described in the results, the solid mixtures contained carbonates. Potassium bicarbonate and sodium carbonate hydrate ($Na_2CO_3 \cdot mH_2O$, m = 1, 5, 7, 10) [40,41] have low melting temperatures (the endothermic peak near 160 °C in the DTA pattern matches the temperature range of potassium bicarbonate: 100–200 °C). Carbonates and chlorides, moreover, may structure eutectic systems [42,43]. The melting point of metallic Pb is approximately 300 °C, which corresponds to an endothermic peak at approximately 360 °C.

Based on a qualitative analysis of XRD for coagulation, equilibrium calculations were conducted using the Equilibrium module of FactSage. According to the calculated results, the system was mainly composed of molten salt, salt crystals ($K_2ZnCl_4$, NaCl), and oxides (FeO and ZnO) at temperature ranges of 200–850 °C. As the temperature decreased from 850 °C to 200 °C, salts precipitated from the molten salt, and the liquid percentage decreased from 50% to 0%. The flue gas with a temperature of 550 °C–850 °C then entered the heat exchanger and exited at <200 °C. The substances in the flue gas underwent a solidification process, which resulted in coagulation.

The microstructure of the coagulum is displayed in Figure 11. In this figure, three structures can be observed: fibers, particles, and blocks. The coagulum contains a considerable amount of carbon, (K, Na)Cl fibers, and (Zn, Fe)O particles. Therefore, a reasonable explanation for the coagulum is proposed as follows. Fine carbon particles have specific surface areas and strong adsorption capacities. The (Zn, Fe)O particles then settle, and (K, Na)Cl crystallizes on the surface (i.e., dendrites and grains). Meanwhile, $ZnCl_2$, carbonate, and Pb dew on the carbon particles. $ZnCl_2$ and (K, Na)Cl form multiple molten-salt systems (block region), thereby increasing the mass of the liquid. Flying particles or fibers then impact the cooling plate and adhere to it. Ultimately, the particles and fibers accumulate in large quantities, forming cakes.

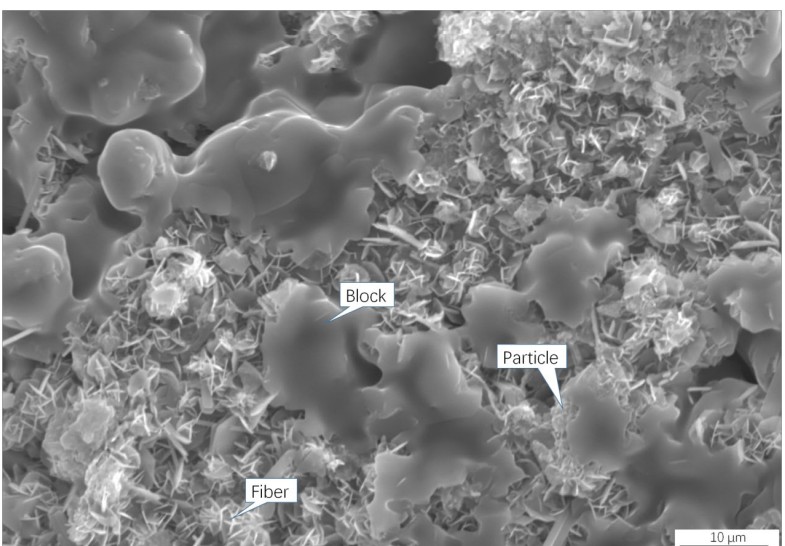

**Figure 11.** Microstructure of the coagulum.

In brief, molten salt, carbonate hydrate, and metallic Pb act as binders; (K, Na)Cl dendrites serve as stiffeners; and (Zn, Fe)O particles act as aggregates, constituting a composite material—coagulum. Among these, the binder is the first requirement for coagulum formation, while the others only increase the strength of the coagulum.

The experimental results demonstrate that the binder originates from substances (with low melting points) containing chlorides, alkali metals, or lead. Metallurgical dust contains one or more of the above three. Caking on the wall of the heat exchanger generally occurs in the processing of metallurgical dust. Considering that inductive heating melts the coagulum, and pulse blasting assists in stripping it, inductive-heating-assisted pulse blasting could be an efficient method to exfoliate the coagulum.

## 5. Conclusions

The findings of the present study, drawn from the experimental results and thermodynamic analyses, can be summarized as follows:

(1)    Some of the oxides in the feedstock were reduced and chloridized to form metals and chlorides. These oxides volatilized into the flue gas at high temperatures. The

produced metals were then re-oxidized, and the chlorides became condensed at lower temperatures, forming a coagulum on the cooling wall of the flue duct.

(2) The coagulum consisted of chlorides ($KCl$, $NaCl$, and $ZnCl_2$), oxides ($ZnO$, $FeO$), and carbon, exhibiting three structures: lumps, fibers, and particles. The liquid phase (eutectic system of $KCl$–$NaCl$–$ZnCl_2$), dendrites ($KCl$, $NaCl$), and particles (($Zn$, $Fe$)$O$) served as binders, stiffeners, and aggregates, respectively, constituting a composite structure.

(3) Liquids, which were essential for the formation of coagulants, originated from the $KCl$–$NaCl$–$ZnCl_2$ eutectic system. The eutectic system played a crucial role in coagulum formation, serving as a binder.

Based on observing the chemical composition, phase constituent, and structure of the coagulum, the formation mechanism of the coagulum was clarified using reaction thermodynamics and phase diagrams, which provided a greater theoretical understanding of concretion formation.

**Author Contributions:** D.W.: methodology, validation, formal analysis, investigation, data curation, writing–original draft, and project administration. S.H.: validation, formal analysis, investigation, data curation, writing–original draft, and project administration. L.W.: validation, formal analysis, data curation, writing–original draft, funding acquisition, writing–review and editing. K.L.: methodology, project administration, and writing–review and editing. H.W.: supervision, project administration, and resources. All authors have read and agreed to the published version of the manuscript.

**Funding:** This research was funded by the National Natural Science Foundation of China, grant number 51774001.

**Data Availability Statement:** Not applicable.

**Conflicts of Interest:** The authors declare no conflict of interest.

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
