# Peer review of "Coupled Thermodynamics and Phase Diagram Analysis of Gas-Duct Concretion Formation in Pyro-Processing Ironmaking and Steelmaking Dust"

_minerals, doi:10.3390/min11101125_

Round 1

Reviewer 1 Report

  • The English writing should be improved.
  • The authors should double check the whole manuscript to remove the typos, e.g., ‘They have may originated’ in line 295.
  • The authors introduced several approaches of dust treatment, e.g., overall utilization and metal recycling. I am expecting some graphic illustration of these methods or a summary using a table.
  • In Figure 1, you may want to demonstrate the different sections and temperature in color.
  • Where is the coagulum in your experiment from, in the paragraph below Figure 1?
  • Please add scale bar to Figure 2.
  • In the paragraph in lines 133-137, please distinguish the capital C from carbon.
  • Double check the title of Table 2.
  • More details about the experiments should be offered, e.g., key procedures of experiments and the precision of the experimental device.
  • A comprehensive flowchart is recommended to show the process of various reactions, including the origin of reaction materials mentioned in the equations of section 4. For example, where is the HCl from?
  • Please indicate the experimental data points obtained in this work in Figures 8 and 9.
  • How the impurities such as FeCl2, CaCl3, and PbCl2 impact the phase diagram in Figures 8-10?

Author Response

(1) The English writing should be improved.

Response: We thank you for your thoughtful and thorough review. The English has been improved by a professional editing service.

(2) The authors should double check the whole manuscript to remove the typos, e.g., ‘They have may originated’ in line 295.

Response: We have carefully checked the entire manuscript and corrected the typos.

(3) The authors introduced several approaches of dust treatment, e.g., overall utilization and metal recycling. I am expecting some graphic illustration of these methods or a summary using a table.

Response: We included graphic illustrations of these methods in the revised manuscript.

(4) In Figure 1, you may want to demonstrate the different sections and temperature in color.

Response: This has been added to the revised manuscript.

(5) Where is the coagulum in your experiment from, in the paragraph below Figure 1?

Response: We have included the location of cogulation in the revised manuscript.

“some substances in the gas coagulate on the cooling wall of heat exchanger (section 3 in Figure 3)” (p. 3)

(6) Please add scale bar to Figure 2.

Response: We have added a scale bar to Figure 2, which is Figure 4 in the revised manuscript.

(7) In the paragraph in lines 133-137, please distinguish the capital C from carbon.

Response: We have replaced the capital C in carbon.

(8) Double check the title of Table 2.

Response: We have corrected the caption of Table 2

(9) More details about the experiments should be offered, e.g., key procedures of experiments and the precision of the experimental device.

Response: In the revised manuscript, we described the procedures of experiments and the precision of the experimental device in detail.

“In the study, the mixtures of metallurgical dusts (such as blast furnace dust and electric arc furnace dust) and other metallurgical solid wastes (for example steel pickling sluddge) were treated in a rotary kiln. The coagulum blocks flue gas flow, which suspends operation at 100–200-day intervals. After maintenance, the next period is started.

We investigated two production cycles which lasted 114 (case 1) and 151 (case 2) days. In each cycle, the feedstock and crude zinc powder were sampled once every 10 days. In a shutdown, five coagulum blocks were obtained from different locations of the heat exchanger along the gas inlet to the outlet. Each block was cut into three pieces perpendicular to the thickness of coagulum, and each piece was cut into two blocks parall to the thickness direction. One block was smoothed and polished. The samples were prepared without a cleaning agent to prevent a reaction, and cleaning was performed using a dryer. Another block was first crushed in a vibration mill and then powdered in a ball grinder for 5 h.

The three powders of the feedstock, coagulum, and crude zinc powder were dried in an oven for 24 h at 110 °C. The dried powders were ground in a ball grinder for 5 h. The chemical compositions of the dried powders were determined by X-ray fluorescence (ARL Advant’X IntelliPower 3600, precision: ± 5%). The phases in the three were analyzed by X-ray diffractometry (Bruker D8 Advance, Cu K). Chemical compounds in the coagulum were determined using a Raman spectrometer (inVia). The thermal properties of the three were analyzed at a heating rate of 10 K/min in an air atmosphere using a thermal analyzer (DTG-60H, precision: ± 1 °C and ± 1%). The morphology of the coagulum block sample was observed using a scanning electron microscope (SEM, JSM-6510), and the compositions of the phases in the coagulum were determined using energy dispersive spectroscopy (EDS, INCA Feature X-Max 20, precision: ±5%).”

(10) A comprehensive flowchart is recommended to show the process of various reactions, including the origin of reaction materials mentioned in the equations of section 4. For example, where is the HCl from?

Response: We have added the description of the process of various reactions in Section 4

“The feedstock consisted of blast furnace dust, electric arc furnace dust, steel pickling sludge, etc. The blast furnace dust contained sodium chloride and potassium chloride, and steel pickling sludge contained ammonium chloride and ferrous chloride. Ammonium chloride and ferrous chloride decomposed in the preheating zone, producing hydrogen chloride, which, in turn, reacted with the oxides. In the high-temperature section, excess carbon in the feedstock leads to an extremely high ratio of partial pressure of the monoxide to that of carbon dioxide. Some oxides were directly reduced, producing metals.

During the flue gas transition from the high temperature to low temperature zones, the reductibility gradually decreased, which oxidized the metals in flue gas.”

(11) Please indicate the experimental data points obtained in this work in Figures 8 and 9.

Response: We have plotted experimental data in Figure 10.

(12) How the impurities such as FeCl2, CaCl2, and PbCl2 impact the phase diagram in Figures 8-10?

Response: We have described the effect of FeCl2, CaCl2, and PbCl2 on the phase diagram in Figures 10

“In addition, Robelin et al [39] show that KCl–NaCl–FeCl2 is similar to KCl–NaCl–ZnCl2, producing a low melting temperature system. The other chlorides, including CaCl2, FeCl2, and PbCl2, further lower the melting temperature of the salt system, improving coagulation formation.”

Reviewer 2 Report

this manuscript needs a lot of editing and a thorough proofread by a native english speaker. There are too many mistakes, typos and editing errors to make it easy to read. The science is sound, but reads like a master students' level project.

Author Response

(1) this manuscript needs a lot of editing and a thorough proofread by a native english speaker. There are too many mistakes, typos and editing errors to make it easy to read. The science is sound, but reads like a master students' level project.

Response: We thank you for your thoughtful and thorough review. We modified the manuscript and substantially improved the English with the help of Editage (https://www.editage.cn/.)

Reviewer 3 Report

Thank you for this paper, which is helpful to the industry, where accretions in off-gas systems are often a concern.

The language of this paper is not yet at publishable level. Some examples:

  • Typo: Digram in title
  • Congealing: do you mean condensing? Accreting?
  • Thermodynamically understanding: thermodynamical (no adverb)
  • page 2 "form steam": steam is only used for water vapour
  • Little attention was given to
  • I will not be able to detect and list all minor errors, so please review the language of the entire paper.

Content:

  • Introduction: inorganic acid leaching: it that only done with sulphuric acid? Not with chloric acid for instance?
  • Chlorination roasting: does this only happen with FeCl2, or do some processes use other reagents? 
  • Materials and methods: the kiln process you focus on, for how much dust is it used? Is this the Waelz kiln process, or different, or broader?
  • Based on the composition and the XRD pattern, can the amount of each phase be estimated? At least what are major and minor phases?
  • Did the DTA analysis use a blank method, can the baseline be corrected?
  • 4.2: being oxidized by feedstock. Not clear what this means
  • 4.3: this section can be shortened. You can use a ternary projection only, and indicate the composition of the observed samples on it
  • To be more complete, it would be good to add an equilibrium calculation for the observed composition of the entire sample and show which phases are present and which are liquid from which temperature. Now there is only basic thermodynamic information and phase diagrams.
  • The amount of samples is very low. How sure can we be that this is representative for the process throughout variations in feedstock, or for the industry? Are other compounds formed at other locations, or other installations, does that depend on the local conditions, the feed mix... To have a more solid scientific work, at least some comparison would be required.

Author Response

(1) Typo: Digram in title

Response: We thank you for your thoughtful and thorough review. We have corrected this in the revised manuscript.

(2) Congealing: do you mean condensing? Accreting?

Response: We mean a mixture of liquids and solids adhere to the wall of the heat exchanger and increase over time.

(3) Thermodynamically understanding: thermodynamical (no adverb)

Response: This has been corrected.

(4) page 2 "form steam": steam is only used for water vapour

Response: This has been corrected.

(5) I will not be able to detect and list all minor errors, so please review the language of the entire paper.

Response: Thank you very much. We have substantially improved the language.

(6) Introduction: inorganic acid leaching: it that only done with sulphuric acid? Not with for instance chloric acid?

Response: We did not find relevant literature focusing chloric acid leaching. The possible reason for this is that, compared to chloric acid, sulphuric acid may decrease the concentration of calcium ions in the lixivium.

(7) Chlorination roasting: does this only happen with FeCl2, or do some processes use other reagents? 

Response: Many substances, such CaCl2 and NH4Cl, may act as chlorinating agents. To process the dust from ironmaking and steelmaking manufacture, researchers have currently only used FeCl2 as the chlorinating agent.

(8) Materials and methods: the kiln process you focus on, for how much dust is it used? Is this the Waelz kiln process, or different, or broader?

Response: We investigated the cogulation in processing the dust from from ironmaking and steelmaking manufacturing in practical production. Tens of thousands of tons of the dust are treated annually in a general kiln process.

(9) Based on the composition and the XRD pattern, can the amount of each phase be estimated? At least what are major and minor phases?

Response: According to the analysis results, we estimated the amount of each phase in the revised manuscript.

“According to qualitative analysis of XRD, average contents of ZnO, KCl, NaCl, ZnCl2, and FeO were approximately 14%, 17%, 9%, 18%, and 11% respectively.”

(10) Did the DTA analysis use a blank method, can the baseline be corrected?

Response: The results showed in the manuscript are based on a reference

(11) 4.2: being oxidized by feedstock. Not clear what this means

Response: We used a specific description in the revised manuscript.

“Away from the high-temperature section, because of suction air and oxidization by the iron oxide in the feedstock”

(12) 4.3: this section can be shortened. You can use a ternary projection only, and indicate the composition of the observed samples on it

Response: We deleted two binary phase diagrams and plotted experimental data on the ternary projection.

(13) To be more complete, it would be good to add an equilibrium calculation for the observed composition of the entire sample and show which phases are present and which are liquid from which temperature. Now there is only basic thermodynamic information and phase diagrams.

Response: We added the equilibrium calculation to explain the cogulation formation.

 “Based on qualitative analysis of XRD for coagulation, equilibrium calculation was conducted using the Equilibrium module of FactSage. According to the calculated results, the system is mainly composed of molten salt, salt crystal (K2ZnCl4, NaCl), and oxides (FeO and ZnO) when the temperature ranges 200–850 ℃. As the temperature decreased from 850 ℃ to 20 0℃, salts precipitate from the molten salt and liquid percentage decreases from 50% to 0%. The flue gas with the temperature of 550–850 ℃ enters the heat excahnger and exists at <200℃. The substances in the flue gas undergo a solidification process, which reults in cogulation.”

(14) The amount of samples is very low. How sure can we be that this is representative for the process throughout variations in feedstock, or for the industry? Are other compounds formed at other locations, or other installations, does that depend on the local conditions, the feed mix... To have a more solid scientific work, at least some comparison would be required.

Response: We have added more experimental data, the samples from different locations, and compared the results of two production cycles.

“In the study, the mixtures of metallurgical dusts (such as blast furnace dust and electric arc furnace dust) and other metallurgical solid wastes (for example steel pickling sluddge) were treated in a rotary kiln. The coagulum blocks flue gas flow, which suspends operation at 100–200-day intervals. After maintenance, the next period is started.

We investigated two production cycles which lasted 114 (case 1) and 151 (case 2) days. In each cycle, the feedstock and crude zinc powder were sampled once every 10 days. In a shutdown, five coagulum blocks were obtained from different locations of the heat exchanger along the gas inlet to the outlet.”

“The thicknesses of the coagulum from the inlet to outlet of the heat exchanger gradually increased, and then decreased. Two production cycles of 114 and 151 days correspond to 7.6% and 6.5% average contents of choloride in the feedstock, while the contents of the other elements have minor differences, which indicates that chloride may play an important role in coagulum formation.”

“A diagram of the KCl–NaCl–ZnCl2 ternary system was produced, and experimental data of the three salts are plotted on the diagram. As shown in Figure 10, the isotherm nearer to the vertex of ZnCl2 corresponds to lower temperatures. Compared to case 2, the locations of the data in case 1 are closer to the vertex of ZnCl2, which indicates that coagultion is formed easily; shorter operation time verified this.”

Round 2

Reviewer 1 Report

The authors have made adequate revisions. Therefore, I recommend it to be published in this fashion.

Author Response

 (1) The authors have made adequate revisions. Therefore, I recommend it to be published in this fashion.

Response: We thank you  for your thoughtful and thorough review, and support.

Reviewer 2 Report

 Once again, the authors should double check the whole manuscript to remove the typos...

line 36 : "In recent years, some researchers have conducted
several experiments and proposed some methods." Utterly useless sentence. Consider giving examples and citing literature.

XRD measurement protocol is not given : what step size, what time per step, how much sample, how where the contents calculated from the XRD pattern, etc... ?

Author Response

(1) Once again, the authors should double check the whole manuscript to remove the typos...

Response: We thank you for your thoughtful and thorough review. We modified the manuscript and substantially improved the English with the help of MDPI Author Services (https://www.mdpi.com/authors/english#English_Editing_Services)

(2) line 36 : "In recent years, some researchers have conducted
several experiments and proposed some methods." Utterly useless sentence. Consider giving examples and citing literature.

Response: We deleted the useless sentence.

(3) XRD measurement protocol is not given : what step size, what time per step, how much sample, how where the contents calculated from the XRD pattern, etc... ?

Response: we added the details and analysis method.

“The phases in the three powders (about a 5 g sample each) were analyzed by X-ray diffractometry (Bruker D8 Advance, Cu K) with a step size of 0.01 degree per 0.3 seconds. Phase percentages were determined by the k-value method using silica as a reference.”

“Based on our XRD experiments using the k-value method, we conducted qualitative calculations. Moreover, equilibrium calculations were performed according to the chemical compositions of the samples. Using these two techniques, the average contents of ZnO, FeO, KCl, NaCl, and ZnCl2 were evaluated, with approximate values of 14%, 11%, 17%, 9%, and 18%, respectively.”
